# The PSA-NCAM-Positive “Immature” Neurons: An Old Discovery Providing New Vistas on Brain Structural Plasticity

**DOI:** 10.3390/cells10102542

**Published:** 2021-09-26

**Authors:** Luca Bonfanti, Tatsunori Seki

**Affiliations:** 1Neuroscience Institute Cavalieri Ottolenghi (NICO), 10043 Orbassano, Italy; 2Department of Veterinary Sciences, University of Turin, 10095 Torino, Italy; 3Department of Histology and Neuroanatomy, Tokyo Medical University, Tokyo 160-8402, Japan; 4Department of Anatomy and Life Structure, Juntendo University Graduate School of Medicine, Tokyo 160-8402, Japan

**Keywords:** brain plasticity, comparative neuroplasticity, adhesion molecules, neuronal markers, human brain, piriform cortex, hippocampus

## Abstract

Studies on brain plasticity have undertaken different roads, tackling a wide range of biological processes: from small synaptic changes affecting the contacts among neurons at the very tip of their processes, to birth, differentiation, and integration of new neurons (adult neurogenesis). Stem cell-driven adult neurogenesis is an exception in the substantially static mammalian brain, yet, it has dominated the research in neurodevelopmental biology during the last thirty years. Studies of comparative neuroplasticity have revealed that neurogenic processes are reduced in large-brained mammals, including humans. On the other hand, large-brained mammals, with respect to rodents, host large populations of special “immature” neurons that are generated prenatally but express immature markers in adulthood. The history of these “immature” neurons started from studies on adhesion molecules carried out at the beginning of the nineties. The identity of these neurons as “stand by” cells “frozen” in a state of immaturity remained un-detected for long time, because of their ill-defined features and because clouded by research ef-forts focused on adult neurogenesis. In this review article, the history of these cells will be reconstructed, and a series of nuances and confounding factors that have hindered the distinction between newly generated and “immature” neurons will be addressed.

## 1. Introduction

Plasticity is an effective mechanism whereby neural elements undergo structural and functional changes, thus enabling the central nervous system (CNS) to complete the development of its neural circuits [1,2,3]. Plastic changes are affected by life experiences at both the young and adult stages [4,5,6,7]. By affecting plasticity, the environment also influences susceptibility to disease and the schedule of development and aging [8,9,10,11]. As such, studies aimed at understanding plasticity are promising for the treatment and prevention of a wide variety of neurological disorders (e.g., dementia and other neurodegenerative diseases [12,13,14,15]). Despite the vast amounts of data produced across the years by neurobiological research, many aspects of brain plasticity remain obscure, especially as to whether and how plasticity is an extension of “embryonic features” or “immaturity”. In addition, it is more and more evident that brain plasticity can remarkably vary as to its type, location, and extension among mammalian species [16,17,18,19,20]. Currently, it is an open question which types of plasticity, and time courses, are sufficiently conserved between model systems and humans as to be relevant to human health [20,21].

The history of CNS structural plasticity investigations started with studies on synaptic plasticity and critical periods (reviewed in [1,4,5]), then flourished with the re-discovery of adult neurogenesis [7,13,22,23,24]. In the complexity of such heterogeneity, cell markers are a useful tool to study plasticity at a cellular level. Across the years, different markers were introduced to detect a wide range of biological processes, from stem cell identity to cell division and from cell migration to neuronal maturation (e.g., doublecortin, nestin, tubulinJ-1, SOX, PAX, ASCL1 genes, NeuN, CNGA3, NeuroD, Calretinin, as well as several markers of cell proliferation; reviewed in [25,26,27]). One of the first molecules used in studies on CNS structural plasticity, including adult neurogenesis, was the polysialylated form of the neural cell adhesion molecule (PSA-NCAM) [28,29,30] since its expression allows a wide range of neuronal changes, reveals cell immaturity, and provides the complete visualization of the cells and their processes (see below). Thirty years after the introduction of this marker, its detection can still provide insights into apparently unsolved controversies on human adult neurogenesis.

### 1.1. PSA-NCAM and Its Multifaceted Properties

PSA-NCAM was discovered in the early 1980s and was sometimes referred to as the embryonic form of NCAM [31]. NCAM is a member of the immunoglobulin superfamily and participates in cell-cell and cell-extracellular matrix adhesion [32,33,34]. The extracellular domain of the major isoforms of NCAM (NCAM180, NCAM-140 and NCAM-120) is composed of five Ig domains followed by two fibronectin type II repeats, and the 5th Ig-domain of NCAM harbors attachment sites for polysialic acid (PSA) or the α2,8-linked linear homopolymer of N-acetylneuraminic acid (Figure 1A). NCAM is heavily polysialylated by two Golgi polysialyltransferase, PST (ST8SiaII), and STX (ST8SiaIV). NCAM has negatively charged long chains of sialic acid with a 4-800 degree of polymerization that occupies an enormous volume around the NCAM peptides and has a steric effect on cell-cell interaction so that PSA disrupts cell-cell adhesion by NCAM and other cell adhesion molecules [31,35,36]. This means that PSA serves as a modulator of cell-cell interactions.

The expression of PSA attached to NCAM is developmentally regulated. Western blotting analysis revealed that PSA occurs during the embryonic and early postnatal brain development but not in the adult brain. Therefore, in the 1980s, many studies on PSA-NCAM focused on the developing brain and demonstrated that PSA-NCAM plays an important role in neuronal development, such as cell migration [37,38], neurite outgrowth [39], neural pathfinding [40,41], and synaptogenesis [31,35,36,42,43].

### 1.2. Heterogeneity of Brain Plasticity and Related Ill-Defined Issues: Cell Markers and Neuronal Immaturity

Decades of research in neurodevelopmental biology have revealed how CNS plastic processes can be heterogeneous under different profiles. At least three main types of structural plasticity are recognized: (i) synaptic plasticity, including formation, elimination, and functional modulation of synaptic contacts [4]; (ii) adult neurogenesis, intended as a lifelong stem cell-driven formation of new neurons [13,24]; (iii) the recently discovered, non-newly generated, “immature” neurons as a form of “neurogenesis without division” (reviewed in [44,45]; see below). Several aspects of these forms of plasticity, including their occurrence, location, extension, and amount, can remarkably vary among mammals at different ages [20]. Current studies reveal that this landscape can be even more complex due to a possible mix/overlapping of neurogenic and non-neurogenic processes, which, in this case, strictly depend on different species and ages [46,47,48].

In this complex topic, one of the highest challenges is that of recognizing specific cell populations and/or biological processes acting on the dynamics of plasticity by using “specific” markers. Nevertheless, some recurrent pitfalls in their use are underestimated. Besides the undoubted usefulness of many cell markers, some of those employed in neurodevelopmental research are hardly specific for single events. First, they can vary depending on anatomical location, animal species, and age, with their generalization possibly leading to erroneous conclusions [49] (Figure 1B). Second, a certain vagueness can depend on their expression as a gradient undergoing changes with time, rather than representing an absolute and constant feature of the cell. Conversely, when the detection of cells that are immunoreactive for the same molecular marker(s) do not necessarily identify a specific type of cell or biological process, this information must be contextualized. This was the case of PSA-NCAM (and, later, of the cytoskeletal protein doublecortin (DCX); see below), which has been historically linked to cell migration and neurogenesis [50], yet is also expressed by a wide range of non-migrating, non-dividing cell populations involved in other forms of plasticity [31,35,36,39,42,43,50,51,52] (Figure 1B).

Why did the heterogeneity of PSA-NCAM as a marker remain undetected for such a long time? The answer is in the timetable of discoveries: soon after the first observations on PSA-NCAM distribution and emerging functions, the landscape of brain plasticity was enriched (and complicated) by the re-discovery of adult neurogenesis, a biological process in which polysialylation is heavily involved [22,53,54]. In these studies, the molecule was often associated with neurogenesis and cell migration, being considered an “easy and reliable marker” for their detection. A notable exception to this vision is represented by a population of non-newly generated “immature” neurons [49]. In the following paragraph, the subsequent steps leading to the current definition of “immature” neurons will be summarized, with special reference to confounding elements coming from their similarities with adult neurogenesis. 

## 2. A Brief History of “Immature” Neurons

The history of “immature” neurons was not a straight one (Figure 2, blue strip). The first step occurred at the beginning of the nineties, in the context of studies on adhesion molecules in the nervous system. At that time, several laboratories were showing the existence of molecules affecting cell-to-cell contacts in the brain by modulating adhesion [32,33,34]. This research field was (and still is) considered of paramount importance in the perspective of modulating neuronal, neuro-glial, and synaptic plasticity [36,52,55,56,57].

In 1991, PSA-NCAM-expressing cells were unexpectedly found in two adult brain regions of rats: an inner part of the granule cell layer in the dentate gyrus, called the subgranular zone (SGZ [29]), and layer II of the piriform cortex [28]. Soon after, two reports performing the complete mapping of PSA-NCAM in the adult rat CNS extended the number of regions expressing this “embryonic” isoform of MCAM (including the olfactory bulb, hypothalamus, mesencephalic central grey, ependyma of the central canal, and superficial laminae of the dorsal horn in the spinal cord [57,58]. These studies used different anti-PSA antibodies to confirm the presence of PSA-NCAM in the piriform cortex and hippocampus.

Starting from the years 1993–1995, the field of neurodevelopmental biology was shaken by the seminal papers on adult neurogenesis [22,23,59]. Adult neurogenesis in the dentate gyrus (Altman and Das, 1965) [60] and the forebrain subventricular zone-olfactory bulb [61] was originally discovered in the sixties by Joseph Altman by use of ^3^H-thymidine autoradiography, and around 1990, a few groups studied adult neurogenesis still using the autoradiography in the rodent dentate gyrus [62,63]. However, detecting ^3^H-thymidine-labeled newborn cells in autoradiography was a complex task, and only the nuclei were visualized in the proliferating cells. In that same period, studies with a combination of 5-bromo-2′-deoxyuridine (BrdU) pulse-chase experiments and PSA-NCAM immunohistochemistry demonstrated for the first time that PSA-NCAM was expressed by newly generated neurons in the dentate gyrus [22,64]. In the other adult neurogenic region, the forebrain subventricular zone, PSA-NCAM was reported to be expressed by neuroblasts migrating in the rostral migratory stream and directed to the olfactory bulb [65,66]. The PSA-NCAM immunohistochemistry allowed the visualization of the entire detailed shape of newly generated neurons, including the dendrites and axons. Since then, a gold rush for studying the cellular/molecular mechanisms of stem cell-driven adult neurogenesis burst in late 1990 [67,68,69,70,71] and substantially dominated the field for several years (Figure 2, orange stripe), leading to forgetting of PSA-NCAM-positive (PSA-NCAM+) paleocortex neurons. All of these studies proposed PSA-NCAM as a proxy molecule marker for newly generated and developing neurons in the subventricular zone and dentate gyrus of rodents.

This conclusion was supported by data in rodents and was logical since PSA-NCAM is expressed by developing neurons in the adult brain, similar to the embryonic and early postnatal brain. In fact, PSA-NCAM in the adult neurogenic regions was reported to regulate cell migration [37,72,73], differentiation [74,75], neural pathfinding and synaptogenesis [73,76], and neuro-glial plasticity [30,55]. However, it was not easy to assume the function of PSA-NCAM in non-neurogenic regions such as the piriform cortex. At this location, PSA-NCAM expression is associated with small- and medium-sized neurons, mainly in layer II, and these cells extend the dendritic processes into layer I, which receives afferent fibers from the olfactory bulb. The PSA-NCAM-expressing neurons also give rise to axons that are incorporated into the sub adjacent white matter.

These morphological features suggest that PSA-NCAM-expressing neurons might be integrated into neuronal circuits of olfactory information. In the olfactory bulb, neuronal circuits are always modified by the addition of newly generated granule cells; therefore, it was presumed that in the piriform cortex, the reorganization of PSA-NCAM-expressing neurons according to the changing input from the olfactory bulb might be required [28]. Nevertheless, at that time, at least three confounding elements made it difficult to focus on the “immature” neuron identity of these cells (see below for further definition). First, most studies focused on the search for more and more sites of postnatal neurogenesis, as well as on neural stem cells, their “niches”, possible modulation, and the potential for brain repair (reviewed in [7,13,14,24,79]). Second, some cell markers for brain structural plasticity, cell migration, and neuronal shaping (e.g., PSA-NCAM and doublecortin, DCX), initially considered as reliable tools for identifying neurogenesis, only later turned out to also be present in populations of prenatally generated (non-dividing) neurons, thus losing their reliability as indicators of neurogenesis (see [49] and below). Third, the existence of actual interspecies differences among mammals concerning the occurrence, type, location, and amount of structural plasticity, started to reveal unexpected twists when non-rodent mammals (including humans) were compared to laboratory rodents [16,17,18,19,47,80]. Nowadays, some of these differences are definitively accepted [16,17,18,49,81]. Nevertheless, in the past, in the absence of such information, results found to be not congruent between mice and other mammals were erroneously discussed (Figure 2, grey strip). 

Only seventeen years after the first description of the PSA-NCAM+ piriform cortex neurons, a Spanish group coordinated by Juan Nacher, using BrdU pulse-chase experiments in rats of different ages (from fetal life to adulthood), clearly demonstrated that neurons in the piriform cortex layer II are generated prenatally, retaining an immature neuronal phenotype into adulthood [82] (Figure 2). At this point, the following questions and hypotheses were generated: do “immature” neurons change numbers with increasing ages? Do they progressively mature or die? In the case where they mature, will they integrate into the cortical circuits as functional elements? Do they represent a “reserve” of young, undifferentiated neurons in the adult cerebral cortex? Which is their physiological role [44,45]?

Some answers to these questions came only recently from studies carried out in Austria by the group led by Sebastien Couillard-Després, using a DCX-Cre-ERT2/Flox-EGFP transgenic mouse, in which GFP is permanently expressed in DCX-positive cells, and in their progeny, following tamoxifen administration [77] (Figure 2). This study confirmed that immature cortical neurons in the mouse piriform cortex do not die with age progression; on the contrary, most of them mature as glutamatergic neurons. After characterization of these cells with patch-clamp experiments, the same research group showed that some elements with more ramified dendritic arborization (complex cells) could functionally integrate into the pre-existing piriform cortex network [83].

Hence, at least in a restricted paleocortex region of rodents, a “reservoir” of undifferentiated cells can provide new functional neurons during adulthood, without cell division or occurrence of stem/progenitor cells. This is because the piriform and entorhinal cortices are interposed between the olfactory bulb and the hippocampus, namely two sites of active neurogenesis. This led some researchers to figure out that these paleocortex regions might also be neurogenic [84]. Nevertheless, data from several laboratories converge to deny such possibility, at least for most of the PSA-NCAM/DCX-positive cells [46,82,85,86,87].

At this point, some remaining questions were: why are these “immature” neurons restricted to the piriform and entorhinal cortices? Are there interspecies differences concerning their occurrence and distribution in mammals? Across the years, some reports observed the occurrence of cortical layer II immature neurons in the neocortex of different mammals [87,88,89,90,91], and, more recently, in subcortical regions, such as the amygdala [46,47,90,92] and claustrum [46], suggesting that results obtained in laboratory rodents substantially differ from other animal species. These comparative analyses started to reveal a heterogeneous interspecific distribution of “immature” cells. In a systematic study carried out on twelve mammalian species, belonging to eight orders with widely different brain sizes, gyrencephaly, lifespan, and ecological niche had remarkable phylogenetic variations, with the “immature” neurons being consistently present in the entire neocortex of large-brained species [78] (Figure 3C). After counting the layer II cortical immature neurons in 84 brains to obtain a linear density (number of cells/mm), a difference of one order of magnitude was found between mouse and cat, with a high amount of cortical “immature” neurons also detected in sheep and chimpanzees [78] (Figure 3D). The comparative data on anatomical extension and the relative amounts of these cells in mammals indicate that they might have been chosen by evolution as a “reservoir of young neurons”, representing a form of “neurogenesis without division” into the adult cerebral cortex circuit of large-brained species [78,93]. Although the abovementioned results raise exciting questions and opportunities under the evolutionary and translational profiles, the issue of “immature” neurons remains largely unexplored and quite ill-defined. In the following paragraphs, the reasons for such unsolved gaps will be addressed.

## 3. The “Immature” Neurons: Possible Heterogeneity and Problems for Their Definition

Most knowledge currently available on these cells is focused on the population of “non-newly generated” cortical neurons existing in rodents, namely, the layer II piriform cortex PSA-NCAM+/DCX+ neurons [44,45,77,82,83] (Figure 3A). In addition to some information obtained on their fate and possible functional integration [77,83], we know for sure that most of them (if not all) are generated prenatally and are already in place into the cortical layer II after the birth of the animal [82,86]. For obvious reasons linked to technical and ethical difficulties encountered in studying larger, more gyrencephalic brains, we know less about neocortical “immature” neurons. Yet, their embryonic origin has been confirmed for some non-rodent mammals, including the highly gyrencephalic cat and sheep [46,87]. On these bases, the cortical “immature” neurons might be considered as a relatively homogeneous population of dormant, non-dividing cells representing a different process of “neurogenesis without division” compared to stem cell-driven neurogenesis [44,45,93]. This feature seems well preserved in all mammalian species studied so far, wherein no double staining with local cell proliferation markers (e.g., Ki-67 antigen) was ever detectable in adult brains [78].

The results obtained in the cerebral cortex would suggest a rather sharp definition of “immature” neurons as non-newly generated, prenatally generated cells. Nevertheless, the situation appears more complex and poorly defined since other populations of DCX+/PSA-NCAM+ cells do occur in various non-neurogenic regions of the mammalian CNS, possibly falling into the definition of “immature” neurons (Figure 3B). Among them, some of the neuronal populations in the amygdala and subcortical white matter [46,47,48,92,94,95]. Yet, it is far from easy to classify such cell populations since they might correspond to/coexist with “non-canonical neurogenic processes”, in which some neurons can be generated out of the canonical stem cell niches, though not reaching the circuit integration outcomes documented in the well-established neurogenic sites [96,97]. For technical and ethical reasons, it is far from easy to study these immature cells with the aim of associating them (or exclude their association) with neurogenic processes (see below). Knowledge of these putative immature neurons is currently very limited. Nevertheless, present data clearly indicate that this kind of cell could be associated with functions linked to higher-order cognitive abilities (neocortex), emotions, and conscience (amygdala and claustrum) that acquire special importance in primate and human evolution (discussed in [46,78]). In all these regions, the immature neurons might represent a possibility to integrate new functional neurons throughout life in the absence of cell division/neural stem cell niches [93]. There are indications that non-cortical immature cell populations vary remarkably as to their occurrence, location, and amount, depending on the animal species. For this reason, systematic comparative studies are needed to describe differences with special reference to large-brained mammals. 

### 3.1. Immature Neurons and Neurogenesis

As outlined in the previous section and in Figure 2, the identification of “immature” neurons was slowed by confusion regarding the existence (or co-existence) of these cells and adult neurogenic events. There are common elements shared by these two biological processes: (i) both can lead to the addition of new functional neurons in pre-existing neural circuits, either with or without cell division [93], at least in the mouse piriform cortex [77,83]; (ii) in some phases of both processes, all “young” neurons share the same markers (e.g., PSA-NCAM and DCX) [49]. 

As introduced above, around the year 2000, another important marker for brain structural plasticity was introduced: the cytoskeletal protein DCX [98,99,100] (Figure 2). Since DCX was abundantly expressed within neurogenic niches, and often co-expressed with PSA-NCAM, and strictly associated with neuroblasts, it was considered a marker for newly born neurons [50]. On this basis, later, every brain region/cell population expressing DCX and/or PSA-NCAM was often interpreted as a source of neurogenesis. Nevertheless, we now know that many CNS regions/neuronal populations expressing PSA-NCAM and DCX are not associated with adult neurogenesis since no cell division/stem cell niche can be found: the piriform and entorhinal cortex [28,58,82], the hypothalamic supraoptic nucleus (Theodosis et al., 1991) [30], the spinal cord dorsal horns [28,58] and central canal [51], and some populations of cortical interneurons [101,102]. 

By using the currently available experimental approaches to detect neurogenesis in vivo (e.g., local cell proliferation markers, such as Ki-67 antigen, PH3, or injected thymidine analogs that can be revealed by antibodies, such as BrdU, EdU), it is extremely difficult to determine when the cells have undergone their last division, especially if a long period of time has elapsed since then. Considering that “immature” neurons are especially abundant in large-brained mammals with respect to laboratory rodents [19,46,49], evident technical and ethical limits do exist in performing this kind of experiment. This is the case of the adult human hippocampus, wherein some studies claim the existence of neurogenesis based on the detection of DCX+/PSA-NCAM+ cells [103,104] in the absence or very low levels of neuronal cell division [105,106,107].

### 3.2. PSA-NCAM Expressing Neurons in the Human Hippocampus

In 1998, a BrdU-labeling study carried out in five adult humans [69] reported the existence of BrdU+ nuclei in the GCL, but subsequent BrdU analyses in humans have not been performed because of obvious technical and ethical reasons, including the toxicity of BrdU (deeper technical considerations in [108]). Most research on adult neurogenesis in the human hippocampus has continued without BrdU analysis by using immature neuronal markers such as DCX and PSA-NCAM [109,110,111,112,113]. Since 2018, there has been a debate whether the level of adult neurogenesis in the adult human hippocampus is very low (undetectable) [105] or persists [103,104]. A similar debate on human adult neurogenesis has continued in the subventricular zone of the forebrain-olfactory bulb since 2004: undetectable [114,115,116] or persisting [117]. In these debates, little attention has been paid to the possibility that “immature neurons” might also exist in previously active neurogenic sites. Since the first studies carried out in the 1990s, it was evident that both the distribution pattern and the age-related change of PSA-NCAM-expressing neurons in the adult dentate gyrus are different when rodents and humans are compared. Meanwhile, in rodent dentate gyrus, PSA-NCAM expressing neurons are restricted to the narrow band of the SGZ [22], are newly generated (as shown by BrdU incorporation analysis [22,67]), and decline with age [64]. In humans, the PSA-NCAM expressing neurons are present in a wider area, including the hilus, and persist in aged individuals [109,110,111]. 

A recent study using a combination of immature neuronal markers (DCX, Hu-B) and a proliferating cell marker (Ki-67 antigen) to overcome the inconvenience of not using BrdU shed light on the possible difference between adult-born neurons and “immature” neurons in the human hippocampus [107,118]. In rodents, a substantial population of proliferating neuronal progenitors simultaneously expressed these molecular markers during neuronal differentiation processes [119]. On the other hand, the analysis in humans [107] demonstrated that the level of proliferation of neuronal progenitor cells is very low, in contrast with a dense population of PSA-NCAM-expressing cells within and out of the GCL (Figure 4), as previously described in studies using only the PSA-NCAM marker [109,110,111]. Accordingly, low levels of cell division (which also include substantial rates of glial cell proliferation), detected and quantified with different methods, independently from the interpretations by different Authors, are a common outcome of most studies carried out on the human hippocampus [103,104,105,106].

How does the low level of neuronal production yield a dense population of immature marker-expressing neurons? To address this question, the following facts should be noted: (i) the period of immature state/maturation of newly generated neurons is much longer in non-human primates than in rodents, suggesting that a small number of proliferating progenitor cells can yield more immature neurons [120,121,122,123]; (ii) as we mentioned above, immature marker-expressing neurons occur in the adult brain of mammals (including non-human and human primates) independently of adult neurogenesis [46,49,78,82,93,124,125,126]; (iii) mature granule cells are able to re-express immature neuronal markers by dematuration [127,128].

Based on these facts, a hypothesis has been proposed that immature neuronal marker-expressing neurons in the human dentate gyrus are derived from (1) adult-born neurons with prolonged immature neuronal marker expression, (2) prenatally born immature neuronal marker-expressing neurons, and (3) immature neuronal marker-re-expressing neurons [118].

Open questions based on this hypothesis are: does there exist a mixed population of newly generated and non-newly generated (“immature”) neurons in the human dentate gyrus? Are some of these neurons generated in postnatal life yet remain (e.g., during adolescence)? Which is the extent of the differences between rodents and humans? Further comprehensive studies of immature neuronal marker-expressing neurons in the adult brain, regardless of whether they are newly generated or non-newly generated neurons, are needed to understand hippocampal plasticity, including adult neurogenesis and its time course. Most of all, there is a need for markers to reconstruct the whole life of the cells after their last mitotic division. Finally, we need further knowledge for translating time science to enhance the interpretation of findings from animal model-based studies [20,21].

## 4. Open Questions and Future Perspectives

The undefined nature of the “immature” neurons and their heterogeneity, along with a scarce number of studies published on this topic, leave a great number of questions still open concerning their physiological role, possible modulation (by environmental cues and/or pathological states), and interspecies differences. Most of all, we still do not know the cellular and molecular mechanism(s) allowing these neurons to stop their differentiation/maturation process at a specific time point, as well as those inducing them to re-start maturation. PSA-NCAM, as a modulator of structural plasticity, can be involved in most of the abovementioned aspects. In a recent report, this molecule is important in the maturation process of mouse cortical immature neurons since the removal of PSA accelerates the final development of these cells [126]. Most importantly, since PSA-NCAM is considered to be a potential target to facilitate repair/regeneration after CNS injury and disease [129], its expression/modulation should be investigated in studies asking whether the immature neurons can be activated in experimental conditions and disease models. 

Another important, unsolved issue in this emerging research field is that of scientific terminology. Many different names have been used by different Authors to indicate both the morphological features of the cells (e.g., tangled cells, complex cells, type 1 and type 2 cells, semilunar cells, extraverted neurons) and the nature of the biological processes (e.g., immature neurons, non-newly generated immature neurons, stand by neurons, dormant neuronal precursors, neurogenesis without division, immature neuronal marker-expressing neurons) [28,44,45,46,51,77,82,83,89,93,118]. The rapid achievements recently obtained on the “immature” neuron topic will require the establishment of new terminology. This is an open issue that cannot be solved in the present review article, and that will require a consensus among scientists working in the field.

Thirty years after the discovery of the PSA-NCAM+ cells in the piriform cortex of rodents [28,58], and after passing through twisted paths and confounding elements, most questions on the “immature” neuron identity and functional significance are still unanswered. The complete mapping of their occurrence, distribution, and amounts in different mammals is ongoing, and most information concerning their presence (and nature) in subcortical regions and white matter is still lacking. Most of all, we need experiments addressing their possible modulation in different environmental and lesion/pathology conditions, as well as insights on their possible role in the maturation/sculpting of neuronal circuits throughout life.

## Figures and Tables

**Figure 1 cells-10-02542-f001:**
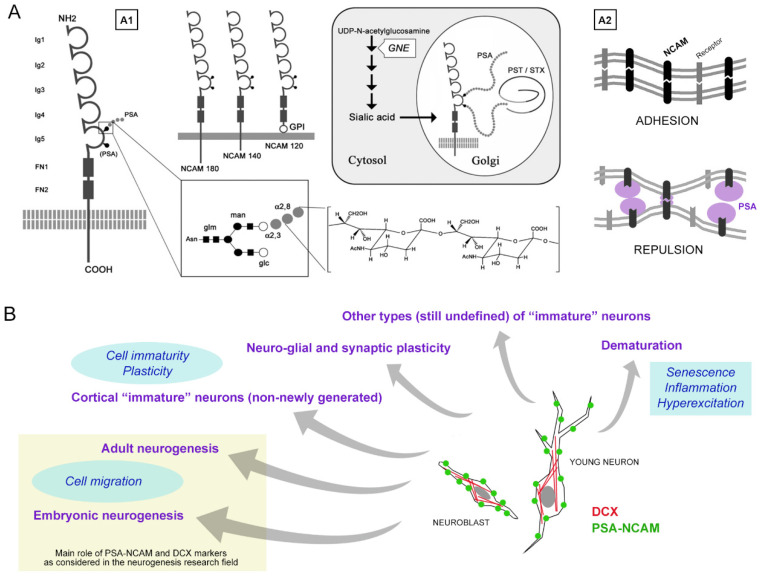
PSA-NCAM molecule and its association with different biological processes. A. Structure (**A1**) and function (**A2**) of NCAM and PSA-NCAM. Polysialic acid (PSA) is a linear homopolymer of a 2-8-linked *N*-acetylneuraminic acid containing from about 8 to over 100 monomers (**A1**). The PSA chains form a large molecule with steric properties due to a high density of negative charges (represented as purple ovals in **A2**), thus allowing modulation of cell adhesion/repulsion. B. Doublecortin (DCX) and PSA-NCAM are frequently co-expressed in cell populations associated with biological processes spanning a very wide spectrum, from neurogenesis to senescence (in contrast with a reductive vision limited to cell genesis and migration, which is often reported in the field of neurogenesis/adult neurogenesis; yellow rectangle). Hence, some plastic cell populations sharing the same markers can remarkably vary in their amount and distribution depending on the brain region or animal species. NCAM, Neural Cell Adhesion Molecule; PSA, polysialic acid; Ig, immunoglobulin; FN1-2, fibronectin type III repeats; GPI, glycosylphosphatidylinositol anchor; PST and STX, polysialyltransferases; GNE, 2-epimerase/Nacetylmannosamine-Kinase. Modified and reproduced with permission from Prog. Neurobiol., Elsevier [53] (**A**), and *Int. J. Mol. Sci.*, *MDPI* [20] (**B**).

**Figure 2 cells-10-02542-f002:**
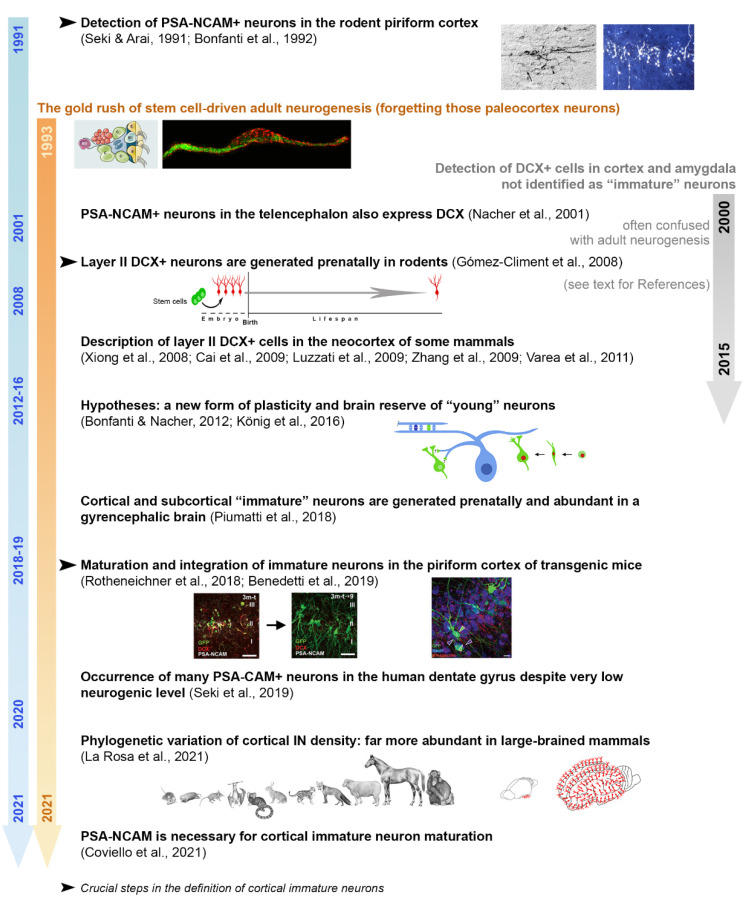
Brief history of the discoveries leading to the current (provisional) definition of non-newly generated “immature” neurons in the last thirty years (blue stripe). The layer II cortical PSA-NCAM^+^ neurons were first described at the beginning of the nineties in the piriform cortex of rodents; then, only in 2008, it was shown that they are generated prenatally. Nevertheless, these cells did not attract attention for a long time since those were years dominated by intensive research on neural stem cells and adult neurogenesis (orange stripe). During this time, several studies found PSA-NCAM^+^/DCX^+^ neurons out of the neurogenic niches, often confusing them with newly generated cells (grey stripe). New information was obtained recently, between 2018 and 2021: we now know that these undifferentiated neurons can mature through adulthood and integrate into pre-existing circuits. Finally, the discovery that they are far more abundant in large-brained mammals (largely extending in the neocortex and subcortical regions) with respect to rodents is drawing attention to this neuronal population as a potential brain reserve of “young” neurons within regions of the human brain not endowed with neural stem cells and adult neurogenesis. PSA-NCAM, the polysialylated form of the neural cell adhesion molecule; DCX, doblecortin. Images modified and reproduced with permission from Anat Embryol., Springer Nature; Prog. Neurobiol., Elsevier; Front. Neurosci., Frontiers Media; Cereb. Cortex, Oxford University Press; eLife, Elife Sciences Publications [44,49,77,78].

**Figure 3 cells-10-02542-f003:**
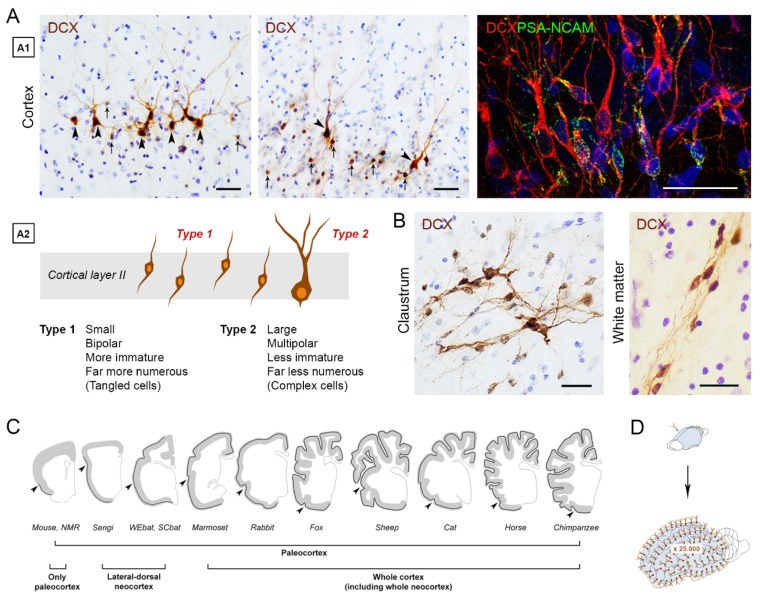
Some features concerning the morphology (**A**), regional distribution (**A**–**C**), and phylogenetic variation (**D**) of the “immature” neurons. Cortical immature neurons can be found in the paleocortex (piriform and entorhinal cortices), neocortex (depending on the species, see **C**), and subcortical regions (**B**, bottom right), including the claustrum, amygdala, and white matter (e.g., external capsule). At present, we have a better understanding of the immature cortical neurons (**A1**): two main cell types have been described (**A2**), corresponding to different maturational stages (small, bipolar, highly undifferentiated cells, and larger, multipolar “complex” cells at further stages of maturation; see text). Scale bars: 30 µm. (**C**,**D**) Comparative studies have shown that large, more gyrencefalic brains are highly enriched in cortical “immature” neurons, concerning both their extension in the cortical mantle (black line in **C**) and their absolute amount (**D**, estimation of total number of immature neurons in mouse and chimpanzee neocortex). PSA-NCAM, polysialylated form of neural cell adhesion molecule; DCX, doblecortin; NMR, naked mole rat. WEbat, Wahlberg’s epauletted fruit bat; SCbat, straw-colored fruit bat. Modified and reproduced with permission from J. Neurosci., Soc. Neuroscience [46] (**A**,**B**), and eLife, Elife Sciences Publications [78] (**C**,**D**).

**Figure 4 cells-10-02542-f004:**
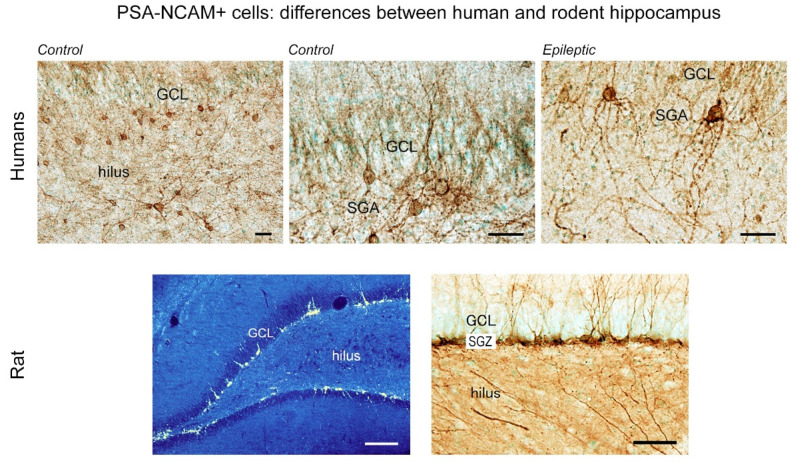
PSA-NCAM expressing cells in the adult human hippocampus and differences with laboratory rodents. In rodents, PSA-NCAM+ cells are confined to the narrow band (subgranular zone, SGZ) below the granule cell layer (GCL), whereas in humans, PSA-NCAM+ cells are distributed in a broadband (subgranular area, SGA) below the GCL, and also in the hilar region. Higher magnification images in the SGA depict some PSA-NCAM+ cells with morphology similar to immature rat neurons in a control patient and aberrant cells with multi-basal dendrites in an epileptic patient. The origin and nature of immature cells in the human hippocampus are still a matter of discussion. Scale bar: 50 µm. Modified and reproduced with permission from Sci. Rep., Springer Nature [107].

## Data Availability

Not applicable.

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
