# Peer review of "The PSA-NCAM-Positive “Immature” Neurons: An Old Discovery Providing New Vistas on Brain Structural Plasticity"

_cells, 2021, doi:10.3390/cells10102542_

Round 1

Reviewer 1 Report

The article is devoted to the history of PSA-NCAM cells and a series of nuances and confounding factors that have hindered the distinction between newly generated and
“immature” neurons.

Overall an in-depth scholarly treatise of currently important issues. However, several  issues need to be addressed:

  1. It seems logical to supplement the information about neuroblasts in the subventricular zone and make a generalizing original scheme for paragraph 3.2.
  2. The figures in the article indicate the receipt of permission to copy information. You should specify in more detail when, by whom the permission was obtained.
  1. There is evidence of the role of PSA-NCAM as a potential target to facilitate repair/regeneration after CNS injury and disease https://pubmed.ncbi.nlm.nih.gov/32898689/.The review would benefit from a data on practical value for PSA-NCAM studying. 

Author Response

The article is devoted to the history of PSA-NCAM cells and a series of nuances and confounding factors that have hindered the distinction between newly generated and “immature” neurons.

Overall an in-depth scholarly treatise of currently important issues.

We thank the Reviewer for his/her positive comment.

However, several  issues need to be addressed:

1) It seems logical to supplement the information about neuroblasts in the subventricular zone and make a generalizing original scheme for paragraph 3.2.

We agree with the Reviewer: the supplement of the information about neuroblasts in the subventricular zone is important logically, and is helpful for the readers. According to the reviewer’s suggestion, we added the information in the text (Lines 297-298).

Furthermore, we tried to make some generalizing original schemes for paragraph 3.2, but we could not, because i) two research groups in the study of the SVZ may recognize different regions as rostral migratory stream, ii) there is difference in the distribution pattern of PSA-NCAM positive cells between the SVZ and SGZ, iii) each groups counts proliferating cells of different type and 4) there is no information about neuroblasts in the human SGZ.

2) The figures in the article indicate the receipt of permission to copy information. You should specify in more detail when, by whom the permission was obtained.

The requested detail have been added in all figures.

3) There is evidence of the role of PSA-NCAM as a potential target to facilitate repair/regeneration after CNS injury and disease https://pubmed.ncbi.nlm.nih.gov/32898689/.The review would benefit from a data on practical value for PSA-NCAM studying. 

A sentence regarding the role of PSA-NCAM as a potential target to facilitate repair/regeneration after CNS injury and disease was added in the last section of the manuscript (Line 354-356). We cited the review article of Saini et al., 2020 (now Ref. n.129) as a good synthesis of this topic.

Nevertheless, at present, due to a lack of studies analysing the possible activation/modulation of the immature neurons in lesion/disease experimental conditions, we think that it is too early to say more.

ALL CHANGES ARE VISIBLE IN RED IN THE REVISED MANUSCRIPT

Reviewer 2 Report

Here, Bonfanti and Seki review the interesting concept of the « immature neurons » and their possible contribution to brain plasticity. This review is well-written and easy to read.

I just have few comments to improve the manuscript :

-Line 51 : Could the authors mention other markers of structural CNS plasticity ? and explain better why PSA-NCAM has been considered as a marker of choice.

-In figure 1, panel C should be removed from this figure since it is cited at the end of the manuscript, line 353. A figure 4 should be created for these pictures.

-In each figure, each panel should be labeled for a better desciption and explanation in the legend. For example, in Figure 1A, there are different images that should be separated (A, B, C).

-In each figure, all abbreviations must be defined in the legend.

-In figure 2, the orange strip must be extended to 2021. References must be added to the orange and grey sentences.

-It would be interesting to discuss why these immature neurons are located in specific regions. Even if the authors ask the question, line 235, they do not tempt to discuss this point.

Author Response

Here, Bonfanti and Seki review the interesting concept of the «immature neurons» and their possible contribution to brain plasticity. This review is well-written and easy to read.

We thank the Reviewer for his/her positive comment.

I just have few comments to improve the manuscript :

-Line 51: Could the authors mention other markers of structural CNS plasticity ? and explain better why PSA-NCAM has been considered as a marker of choice.

The main markers of structural CNS plasticity were listed (lines 46-47).

The reason for PSA-NCAM as a marker of choice was partially described in Line 42-49. We integrated the sentence (lines 49-50).

-In figure 1, panel C should be removed from this figure since it is cited at the end of the manuscript, line 353. A figure 4 should be created for these pictures.

The panel C was removed and a Figure 4 was created ex novo. Following the comments of Reviewer 1, we added here images of PA-NCAM+ cells in the rodent hippocampus.

-In each figure, each panel should be labeled for a better description and explanation in the legend. For example, in Figure 1A, there are different images that should be separated (A, B, C).

We did this in Figure 1A (as A1 and A2 to indicate PSA-NCAM structure and function), and also in Figure 3 and 4.

-In each figure, all abbreviations must be defined in the legend.

Definitions of abbreviations were added.

-In figure 2, the orange strip must be extended to 2021. References must be added to the orange and grey sentences.

We agree. Orange strip has been extended.

As to the References, they are too many to be reported in the figure. They are cited in the text and we indicated this in the Figure.

-It would be interesting to discuss why these immature neurons are located in specific regions. Even if the authors ask the question, line 235, they do not tempt to discuss this point.

This point is extremely interesting, although there are not enough data available at present to answer this question. By following the Referee’s indication, we discussed some hypotheses (added in lines 243-247).

ALL CHANGES ARE VISIBLE IN RED IN THE REVISED MANUSCRIPT